# Discovery and Functional Characterization of Two Regulatory Variants Underlying Lupus Susceptibility at 2p13.1

**DOI:** 10.3390/genes13061016

**Published:** 2022-06-05

**Authors:** Mehdi Fazel-Najafabadi, Harikrishna-Reddy Rallabandi, Manish K. Singh, Guru P. Maiti, Jacqueline Morris, Loren L. Looger, Swapan K. Nath

**Affiliations:** 1Arthritis and Clinical Immunology Research Program, Oklahoma Medical Research Foundation, Oklahoma City, OK 73104, USA; mehdi-fazel@omrf.org (M.F.-N.); harikrishnareddy-rallabandi@omrf.org (H.-R.R.); manishkumar-singh@omrf.org (M.K.S.); guru-maiti@omrf.org (G.P.M.); 2Department of Neurosciences, University of California, San Diego, CA 92121, USA; jfmorris@health.ucsd.edu; 3Howard Hughes Medical Institute, University of California, San Diego, CA 92121, USA

**Keywords:** lupus, Post-GWAS, DGUOK, Luciferase, enhancer

## Abstract

Genome-wide association studies have identified 2p13.1 as a prominent susceptibility locus for systemic lupus erythematosus (SLE)—a complex, multisystem autoimmune disease. However, the identity of underlying causal variant (s) and molecular mechanisms for increasing disease susceptibility are poorly understood. Using meta-analysis (cases = 10,252, controls = 21,604) followed by conditional analysis, bioinformatic annotation, and eQTL and 3D-chromatin interaction analyses, we computationally prioritized potential functional variants and subsequently experimentally validated their effects. Ethnicity-specific meta-analysis revealed striking allele frequency differences between Asian and European ancestries, but with similar odds ratios. We identified 20 genome-wide significant (*p* < 5 × 10^−8^) variants, and conditional analysis pinpointed two potential functional variants, rs6705628 and rs2272165, likely to explain the association. The two SNPs are near *DGUOK*, mitochondrial deoxyguanosine kinase, and its associated antisense RNA *DGUOK-AS1*. Using luciferase reporter gene assays, we found significant cell type- and allele-specific promoter activity at rs6705628 and enhancer activity at rs2272165. This is supported by ChIP-qPCR showing allele-specific binding with three histone marks (H3K27ac, H3K4me3, and H3K4me1), RNA polymerase II (Pol II), transcriptional coactivator p300, CCCTC-binding factor (CTCF), and transcription factor ARID3A. Transcriptome data across 28 immune cell types from Asians showed both SNPs are cell-type-specific but only in B-cells. Splicing QTLs showed strong regulation of *DGUOK-AS1*. Genotype-specific DGOUK protein levels are supported by Western blots. Promoter capture Hi-C data revealed long-range chromatin interactions between rs2272165 and several nearby promoters, including *DGUOK*. Taken together, we provide mechanistic insights into how two noncoding variants underlie SLE risk at the 2p13.1 locus.

## 1. Introduction

Systemic lupus erythematosus (SLE or lupus) is an autoimmune disease characterized by abnormal B- and T-cell responses, production of numerous pathogenic autoantibodies, and immune complex deposition, among other phenomena—leading to myriad clinical manifestations [1,2]. These range from relatively mild manifestations (e.g., skin rash or non-erosive arthritis) to seriously disabling or even life-threatening complications, such as lupus nephritis, neuropsychiatric disorders, and other major organ involvement [3,4,5]. In the US alone, over 200,000 individuals are currently afflicted with active SLE [6], mostly women (~90%), with prevalence 3–5 times higher in individuals of African, Asian, and Hispanic ancestries compared to Caucasian ancestry. In addition, non-Caucasian patients tend to manifest clinical disease earlier and show accelerated and more severe organ damage [7,8,9].

While SLE etiopathogenesis is not fully understood, there is a substantial genetic contribution, as exemplified by frequent familial aggregation [7], high heritability [10], striking monozygotic twin-concordance (a 10-fold increase over dizygotic twins) [11], and large sibling recurrence risk (λs ~ 29) [7]. To date, several large-scale genome-wide association studies (GWAS) and meta-analyses from our lab and others have identified over 100 genome-wide significant (*p* < 5 × 10^−8^) lupus-associated risk loci, mostly linked to single nucleotide polymorphisms (SNPs) located in non-coding regions [12,13]. However, given that credible intervals responsible for GWAS loci typically consist of hundreds of variants with similar statistical support, discerning ‘true’ causal SNP(s) underlying association is difficult [14]—but necessary for a mechanistic understanding of disease development, progression, and involved pathways.

The SLE susceptibility locus at 2p13.1 was first identified in a meta-analysis of Asian (Chinese and Thai) populations [15]. However, despite strong genetic association with SLE, the identities of causal variant(s) and their underlying molecular mechanisms at this locus are unknown. In this study, using extensive bioinformatic analysis followed by diverse in vitro experimental assays across SLE-relevant cell lines, we thoroughly characterized relevant SNPs and molecular involvement at this locus. Our study provides mechanistic insight into how two non-coding variants interact to modulate the expression of their cognate target genes (*DGUOK* and the nearby antisense non-coding RNA *DGUOK-AS1*.) We find that these two SNPs fully explain SLE susceptibility at the 2p13.1 locus.

## 2. Materials and Methods

### 2.1. Meta-Analysis and Conditional Analysis

For this study, we used summary data available from six cohorts, four from Asian [16,17] and two from European [16,17,18] ancestries. For each cohort, sample sizes used for SLE cases and controls are shown in Table 1. This study was approved by the Institutional Review Board of the Oklahoma Medical Research Foundation (OMRF) IRB# 10-23).

To identify associated SNPs from the SLE susceptibility locus at 2p13.1, we extracted all relevant information (SNP, position, allele frequencies, effect sizes). We began with our published Asian cohorts [16,17] and augmented with data from another study [18]. SNP quality control for our initial Asian cohort has been described elsewhere [16,17]. All SNPs in the study were in Hardy-Weinberg equilibrium (*p* > 1 × 10^−6^) and had minor allele frequency >1%. Association analysis for both studies was performed using PLINK v1.9 [19]. Ethnicity-specific meta-analysis for four Asian and two European cohorts was performed by METAL [20], using the weighted inverse variance method—based on regression coefficients (β values), standard errors, and *p*-value—by weighting the effect estimates of individual sample sizes by the inverse of the variance and taking into account effect direction and odds ratio heterogeneity. We then performed conditional analysis on selected SNPs (those with significant *p*-values) to assess whether each SNP was sufficient to explain the association of the entire locus using GCTA software with the GCTA-cojo module [21]. Since the *p*-values for highly associated SNPs are similar in order of magnitude, we began conditional analysis with the most evolutionary conserved SNP. For this, we used “SiPhy-cons,” a measure of evolutionary conservation that measures conservation pressure on single base pairs instead of DNA stretches, as calculated by the PhastCons algorithm implemented in Haploreg [22]. We used a pre-defined threshold of *p* < 5 × 10^−4^ for independent association.

### 2.2. Bioinformatics, eQTL and sQTL Analysis, and 3D-Chromatin Interaction Analysis

We annotated the region with the two selected SNPs with epigenetic marks from the ENCODE GM12878 cell line track. As shown in Figure 1, the two SNPs, especially rs2272165, are located at the peak of the H3K27ac mark, often found near active regulatory regions. (The highlighted region is the interval between the two SNPs, which are in complete linkage disequilibrium).

To evaluate the pathogenicity of the SNPs, we used PredictSNP2 [23], which predicts the effect of nucleotide substitution in any region of the genome. The final consensus score was based on the integration of outputs from CADD, DANN, FATHMM, FunSeq2, and GWAVA.

Regulatory expression quantitative trait locus (eQTL) mapping is a powerful approach to connect disease-associated non-coding variants to gene regulatory mechanisms. To evaluate cell type-specific eQTL and target gene expression, we used ImmuNexUT (Immune Cell Gene Expression Atlas from the University of Tokyo) [24] data, including gene expression and eQTL data from 28 types of immune cells isolated from 10 distinct human immune diseases and healthy donors. When queried, the atlas contained 9852 immune cell samples from 416 donors. We also analyzed splicing quantitative trait loci (sQTLs), i.e., regulation of gene alternative splicing—computed from GTEx RNA-seq datasets.

Chromatin histone marks for the region were extracted from the ENCODE GM12878 cell line track and visualized using the WASHU genome browser [25]. We also extracted chromatin conformation data from a promoter capture HiC (PCHiC) study [26] to determine possible regional chromatin interactions between SNP-containing regions and neighboring genes. We extracted PCHiC data for the SNP-containing region and genes around this region from B-cells and monocytes.

### 2.3. Non-Coding RNA Interaction Analysis

We used the DIANA-LncBase (v3) database [27] to collate experimentally validated miRNA targets of lncRNAs. At the time of our search, DIANA-LncBase v3 included ~250,000 miRNA-lncRNA pairs, collected from 243 distinct human and mouse tissues and cell types. The entries were derived from manual curation of publications and >300,000 publicly available datasets, across 14 different experimental methodologies. We used default settings for our searches.

For prediction of non-miRNA interactions of lncRNAs, we used lncRRIsearch [28] (lncRRIsearch). LncRRIsearch uses RIblast to predict RNA–RNA interactions. We used default settings for our searches.

### 2.4. Expression Level Analysis of DGUOK and DGUOK-AS1 in Four B-lymphocyte Lines

The single-cell expression matrix files containing UMI counts for expressed genes in matrix market exchange format output by the 10× CellRanger pipeline were downloaded from GEO for samples GSM3596321 (Coriell cell line GM12878) and GSM3596320 (Coriell cell line GM18502) [29]. This process was repeated for GEO samples GSM4796271 and GSM4796272, for which sequencing samples and subsequent UMI counts were generated in the same manner as the previous reference but from patient-derived EBV-transformed lymphoblastoid B-cell lines using B95-8 or M81 strains of EBV, respectively [30]. UMI counts for *DGUOK* and *DGUOK-AS1* were retrieved from the matrix files and the number of cells counted for each unique combination of *DGUOK/DGUOK-AS1* expression levels. Cell counts were log-transformed and displayed as a heatmap (see calibration bar below figure panels) for each combination to visualize co-regulation of expression as a function of expression value. The value “−1” represents pairs of expression values for which no cells were found; this was chosen to aid in visualization.

### 2.5. Luciferase Reporter Assay

The Dual-Luciferase Reporter Assay System (Biotium) was used to assess potential allele-specific enhancer/promoter activity of the sequences containing rs67056278 and rs2272165. Briefly, ~400 bp regions surrounding rs67056278 and rs2272165 were cloned into the pGL4.14 vector (for promoter assays) and the pGL4.26 vector (for enhancer assays) (both from Promega). HEK293 (human embryonic kidney cells), U937 (monocyte), and LCL (B-lymphoblastoid) cell lines were cultured and grown up to ~70% confluence, and each plasmid was transiently co-transfected with pGL4.74 (internal control). Enhancer/promoter activity of each construct was measured after 24 h, using the Dual-Luciferase Reporter Assay.

### 2.6. Allele-Specific ChIP–qPCR

Chromatin immunoprecipitation (ChIP) assays were performed using a Magnify ChIP assay kit (Cat No. 492024, Thermo-Fisher, Waltham, MA, USA) according to manufacturer guidelines to test allele-specific binding of the homozygous risk and non-risk genotypes of the rs67056278 and rs2272165 regions to histone marks (H3K27ac, H3K4me1, and H3K4me3), RNA Pol II, P300, CTCF, and ARID3A. Briefly, 1.5–2 × 10^6^ homozygous rs67056278/rs2272165 (both) risk and non-risk haplotype-containing Epstein–Barr virus-transduced B-lymphoblastoid cell lines (from the Coriell collection) were first crosslinked with 1% paraformaldehyde. Cells were thoroughly washed with cold PBS, pelleted, and sonicated (Covaris S1 sonicator, No. E220) in 130 µL of lysis buffer containing protein inhibitor cocktail. Antibodies against the individual histone (H3K27ac, H3K4me1, and H3K4me3) and other DNA-binding proteins (RNA Pol II, P300, and CTCF), as well as a control mouse IgG, were pre-incubated with Dynamag magnetic A+G beads for 2 h at 4 °C. Sheared chromatin–protein complexes were then incubated overnight at 4 °C with mild agitation for immunoprecipitation. DNA was reverse cross-linked by incubating with proteinase K at 55 °C for 25 min and eluted from the immunoprecipitated chromatin complexes. Eluted samples were subjected to real-time qPCR analysis with SYBR Green and primers flanking the rs67056278 and rs2272165 SNP regions, using an Applied Biosystems 7900HT qPCR machine. Experiments were performed in triplicate, and statistical significance was assessed by Student’s *t*-test using GraphPad PRISM software. *p*-value < 0.05 was considered as significant.

### 2.7. Western Blotting

Three lymphoblastoid cell lines (LCLs; 2 homozygous risk/risk and 1 homozygous non-risk/non-risk) from Coriell were used for Western blots. Two non-risk (GM18553, GM18561) and two risk (GM18572, GM18592) lines were lysed in NP40 buffer (50 mM Tris, pH 8.0, 150 mM NaCl, 1% NP40, 1% SDS, 1 mM protease inhibitor cocktail) on ice for 15 min. The prepared lysates were centrifuged at 20,000× *g* at 4 °C for 10 min. Supernatants were collected and used for protein quantification. Further, 100 μg supernatant from each sample (non-risk and risk) was separated on an SDS-PAGE gel and transferred onto a PVDF membrane. The membranes were blocked with 5% BSA in Tris-buffered saline with Tween-20 (TBST) for 1 h at room temperature. Upon completion of incubation, blotting membranes were washed three times with buffer. Next, the blots were incubated with anti-DGUOK (Santa Cruz Biotechnology, SC-376267) and anti-GAPDH (SC-47724) primary antibodies overnight at 4 °C, on a shaker. The next day, blotting membranes were incubated with anti-mouse IgG secondary antibody for 1 h at room temperature on a shaker. Finally, the bands were detected by an ECL-plus Western blotting detection system (Cat No. 32209, Thermo-Fisher, Waltham, MA, USA) following manufacturer guidelines.

## 3. Results

### 3.1. Meta-Analysis of Genome-Wide Association Studies

We first performed two ethnicity-specific meta-analyses across published Asian and European GWAS studies covering the 2p13.1 region (Table 1). The Asian cohorts included individuals from Han Chinese, Malaysian Chinese, and Korean populations, and the European cohorts included individuals from Spanish and European ancestries. The total number of individuals in our meta-analysis was 31,856 (10,252 cases; 21,604 controls). The association signal mapped almost exclusively to the *TET3* region, with particular accumulation at the intergenic region between *TET3* (5′ to it) and *DGUOK* (3′ to it—also the overlapping lncRNA *DGUOK-AS1*) (Figure 2a). Among the 1741 qualified (MAF > 1% in East Asians) SNPs within ~1 megabase (hg19-Chr2: 73,708,783-74,705,126) region, 20 SNPs (18 substitutions and 2 indels) from Asian-specific meta-analysis passed genome-wide significance (*p* < 5 × 10^−8^) (Appendix A). Signal localized to several SNPs near the association peak (Figure 2a). Of these, only rs2272165 was highly conserved (Methods), and we began conditional analysis with this SNP. Conditional analysis localized signal to rs2272165 and rs6705628, with further conditioning on these SNPs essentially removing association signal (Figure 2b). Thus, the association is well explained by these two SNPs.

For these two SNPs, each risk allele occurred at around 20% frequency in the Asian populations, and around 1% in the Europeans, reflecting a profound difference in underlying allelic prevalence. The meta-analysis yielded similar statistics for both SNPs (unsurprising given their close distance—4883 bases—and strong LD between them—r^2^ = 0.99—in Asians.) We analyzed our data on Asian and European ancestries both separately and jointly. The *p*-value and odds ratio (OR) with 95% confidence interval were *p* = 5.56 × 10^−10^ and 0.79 (0.72–0.87) for rs6705628; and *p* = 3.48 × 10^−10^ and 0.79 (0.71–0.86) for rs2272165. Despite the much lower incidence, computed odds ratios were identical (0.91) in European populations. The rs6705628 SNP just achieved statistical significance (*p* = 4.99 × 10^−2^), whereas the rs2272165 SNP just missed it. Taken together, this meta-analysis indicates that these two SNPs (whose effects are difficult to separate due to nearly compete LD) strongly contribute to SLE susceptibility, especially pronounced in Asians. Thus, we moved forward with experiments to determine mechanistic aspects of risk arising at these two SNPs in the 2p13.1 region.

### 3.2. Bioinformatics of the Region

We evaluated the potential pathogenicity of the SNPs by running PredictSNP2 [23] on this genomic region. Several SNPs including rs2272165 were identified as potentially deleterious (Appendix A).

Both SNPs (rs2272165 and rs6705628) are in chromatin that is annotated as quite active (ENCODE data in UCSC Genome Browser). rs6705628 is annotated as an active promoter in GM12878 (B-lymphocyte) cells, and as a strong enhancer in K562 cells (erythroleukemia cells resembling granulocytes). ENCODE project data show very strong histone marking (particularly H3K4me3 and H3K27ac) in both cell lines, and the region is annotated as open chromatin in both, as well. ENCODE ChIP-seq data show a remarkable number of bound transcription factors and chromatin regulators at this locus: JUND, MYC, MAX, KDM5B, SIX5, ZBTB33, ELF1, PHF8, E2F1, TAF1, HMGN3, CHD2, TBP, TFAP2C, CTCF, CREB1, EBTF, ZBTB7A, SP4, IRF1, E2F4, MAX, TEAD4, TBL1XR1, ARID3A, and NR2F2. This base is 100% conserved as the risk C in primates.

rs2272165 is annotated as a strong enhancer in both GM12878 and K562 cells. Histone marking (particularly H3K4me3) is strong in K562 cells, and both cells show this region as open chromatin. ChIP-seq shows binding of MAX, SMARCB1, BHLHE40, MYC, KDM5B, MXI1, E2F6, TBP, NFIC, STAT3, UCTF, and SPI1 to the region. Consensus recognition motifs for MZF1 and RREB1 occur directly adjacent to this base. This base is 100% conserved as the risk G in primates.

Based on 3D-chromatin interaction analysis with the PCHiC data, we found that the rs2272165 region strongly interacts with the promoters of neighboring genes (including both *DGUOK* and *DGUOK-AS1*) in both GM12878 cells and monocytes, as well as all other immune cells together (Figure 1). This suggests a potential regulatory role.

### 3.3. Non-Coding RNA Interactions

The *DGUOK-AS1* antisense RNA presents fascinating connections with the immune system. In addition to targeting its namesake *DGUOK* itself, *DGUOK-AS1* regulates many microRNAs with immune involvement. *DGUOK-AS1* regulates the microRNA miR-204-5p, whose principal downstream target is interleukin-11; in this context, the RNA was found to promote IL-11 secretion, breast cancer cell migration, angiogenesis, and macrophage migration [31,32]. It appears that *DGUOK-AS1* may bind to multiple miRNA species and affect their function. Other experimentally validated miRNA targets of *DGUOK-AS1* (Appendix A) included miR-1-3p, miR-138-5p, miR-148a-3p and miR-148b-3p, miR-151a-3p, miR-653-5p, and miR-876-3p. Remarkably, all interacting miRNAs are primarily annotated as immune modulators.

We next performed a search for non-miRNA targets of *DGUOK-AS1* (Appendix A). The top hits included the inflammatory peptide bradykinin, which is upregulated in SLE, rheumatoid arthritis, and Hashimoto’s thyroiditis [33]; pattern recognition receptor 36, which is involved in the innate immune system [34]; and the lncRNA *Xist*, the principal mediator of X chromosome inactivation in females [35]—*Xist* has been shown to strongly contribute to B-cell modulation and sex bias in SLE and arthritis [36].

### 3.4. Relationship of DGUOK and DGUOK-AS1

The lncRNA *DGUOK-AS1* was discovered through RNA-seq on cervical cancer tissues. The lncRNA consists of two exons, one of which overlaps the penultimate coding exon of *DGUOK* (exon 4 or 6, depending on splice isoform) by 60 bases—this is the origin of the name of the lncRNA. We explored existing single-cell RNA-seq data for B-lymphocytes (Coriell lines GM18502, GM12878, LCL_777_B958, and LCL_777_B958_M81) and discovered that levels of *DGUOK* and *DGUOK-AS1* are indeed largely negatively correlated (Appendix A), as would be expected if *DGUOK-AS1* is targeting *DGUOK* for inhibition (as do most, but not all, lncRNA-target gene pairs). Thus, the two genes appear to be transcriptionally co-regulated, as predicted from genome structure.

### 3.5. eQTL and sQTL Analysis

To better establish the effects of the SNPs on gene expression, we searched publicly available expression quantitative trait locus (eQTL) databases (based on patient-derived primary cells) for the two SNPs. A detailed eQTL analysis across essentially all blood cell types [24] showed that both rs6705628 and rs2272165 were powerful determinants of *DGUOK* expression levels (Figure 3), particularly in naïve B-cells (*p* = 1.48 × 10^−5^ and 1.67 × 10^−5^, respectively), unswitched memory B-cells (USM_B, *p* = 2.41 × 10^−3^ and 2.97 × 10^−3^, respectively), and double-negative effector memory B-cells (DN_B, *p* = 1.01 × 10^−2^ and 1.97 × 10^−2^, respectively). Changes were smaller and less significant in switched memory B-cells (SM_B). We also found that both SNPs are strong sQTLs for *DGUOK-AS1* in multiple tissues: *p* = 2.1 × 10^−24^ for esophageal mucosa and 1.7 × 10^−15^ for skin (Appendix A).

### 3.6. Validation of Allele-Specific Regulatory Effects of SNPs

To assess allele-specific promoter/enhancer effects of the two SNPs, we performed luciferase reporter assays in three cell lines by transient transfection (HEK293 kidney cells, U937 monocytes, and the Coriell lymphoblastoid B-cell line GM18572—“LCL”; see Methods). In HEK293 cells, the rs2272165 locus showed strong promoter and enhancer activity (~6-fold over the empty vector for both)—the risk G SNP lowered activity ~10% over the non-risk A SNP for both—the difference was significant (*p* = 0.009) for enhancer activity (Figure 4a). The rs6705628 locus showed modest promoter and enhancer activity—the risk C SNP increased activity ~40% (*p* = 0.007) for promoter activity. In U937 cells (Figure 4b), the rs2272165 locus showed stronger promoter and enhancer activity (~4-fold and ~8-fold over the empty vector, respectively), and the risk G SNP had a more profound effect on enhancer activity (40% decrease, *p* = 0.03). The rs6705628 locus also showed much stronger promoter and enhancer activity than in HEK293 cells (40% increase and 4x increase over empty vector)—the risk C SNP increased promoter activity ~2.5-fold (*p* = 0.002). In LCL (Figure 4c), both regions were strong, SNP-independent silencers of promoter activity (~2x less than empty vector). Meanwhile, the rs2272165 locus was a strong enhancer (~2.5x over empty vector)—an effect that was completely abolished by the risk G SNP (*p* = 0.002). Taken together, both genomic loci exhibit strong effects on both promoter and enhancer activities—and a single base pair substitution significantly modulates these effects. Overall, the risk genotype at rs2272165 decreased enhancer and promoter activity across all three cell types, whereas the rs6705628 risk genotype increased enhancer and promoter activity across all cell types.

### 3.7. Differential Allele-Specific Binding of Regulatory Proteins to SNP Regions

To better understand the allele-specific regulatory activity of both SNP regions, we first sought to establish the set of histone marks and other proteins interacting with the two loci. To recapitulate in vivo conditions, we measured binding at the regions in their endogenous chromatin state—with proteins expressed at native levels—by making use of genotyped cells (Coriell B-cell lines) homozygous at the two SNPs. From Coriell lines with the appropriate genetic background (Han Chinese in Beijing, CHB HapMap group), we selected (from published genotyping) two homozygous-risk and six homozygous-non-risk lines for verification by TaqMan. Of these verified lines, we chose GM18572 and GM18553, respectively, to represent the risk and non-risk haplotypes.

Using ChIP-grade antibodies (Abcam), we measured in situ allele-specific binding of the histone marks H3K27ac, H3K4me1, and H3K4me3 to each SNP genotype, quantifying binding with ChIP-qPCR (Figure 5). For rs6705628, all three marks bound significantly more in GM18572 (i.e., the risk CC genotype) than in GM18553 (i.e., the non-risk TT genotype) (H3K27ac: *p* = 0.009, ~2.3× increase; H3K4me1: *p* = 0.009, ~5× increase; H3K4me3: *p* = 0.03, ~2× increase). Conversely, for rs2272165, significantly lower binding was seen for all three marks in GM18572 (i.e., the risk GG genotype) than in GM18553 (i.e., the non-risk AA genotype) (H3K27ac: *p* < 0.001, ~2.5× decrease; H3K4me1: *p* < 0.001, ~2.5× decrease; H3K4me3: *p* < 0.001, ~4× decrease). In general, H3K27ac and H3K4me3 mark active promoters, particularly near transcription start sites, whereas H3K4me1 marks active enhancers. The risk rs2272165 genotype decreased promoter and enhancer activity across cell types (Figure 4); thus ChIP-qPCR results are in complete agreement. Similarly, the risk rs6705628 genotype increased promoter activity in monocytes and HEK293 cells; thus, the rs6705628 ChIP-qPCR results are also consistent. Binding of several other proteins was also dramatically increased in the risk rs6705628 genotype (P300: *p* = 0.021, ~7× increase; Pol II: *p* < 0.001, ~4× increase; CTCF: *p* = 0.018, ~4× increase; ARID3A: *p* = 0.007, ~6× increase). It is likely that a complex interaction between RNA polymerase, histone marks, chromatin modulators, and cell type-specific components controls transcription from these loci.

### 3.8. Western Blot Analysis of DGUOK

Given the large allele-specific effects that we observed from the luciferase and ChIP-seq experiments, we sought to determine if the SNP genotypes dictated protein expression levels as well. Using an antibody specific against the DGUOK protein, we performed Western blots on three LCL lysates, from the homozygous rs2272165/rs6705628 risk/risk genotype (GM18572 and GM18555) and the homozygous non-risk/non-risk genotype (GM18553 and GM18561) (Figure 6). The non-risk GM18553 line showed much higher levels of DGUOK protein expression than the risk 101739 and 101972 lines (~50% and 4.5× higher, respectively). These results are consistent with the luciferase experiments, wherein promoter and enhancer activity were lower for risk than non-risk genotype at all 8 measurements (significantly lower at 3 of them). Thus, the transcriptional effects mediated by these two SNPs translate directly into protein levels—importantly, in relevant white blood cell lines from endogenous chromosomal expression, likely mirroring the in vivo situation.

## 4. Discussion

Here we have performed a meta-analysis of the 2p13.1 lupus susceptibility locus across six cohorts from Asian and European populations. The meta-analysis greatly strengthened and refined the association signal in both ancestries, especially predominant in Asian populations. Subsequent conditional analysis localized the association to two non-coding SNPs in the *DGUOK/DGUOK-AS1* locus. Since the allele effects were found in Asians, we used cell type-specific eQTL data from a Japanese population. These two SNPs have strong eQTL signals from primary B-cells, modulating the expression of *DGUOK* significantly. We carried out a series of in vitro experiments on three immortalized cell lines that demonstrated strong promoter and enhancer activity of the local regions around these two SNPs, with the two SNPs greatly modulating activity. The risk allele (G) at rs2272165 decreased enhancer activity across all three cell types. Western blots verified that this decreased transcriptional activity carried over to protein expression, which was several-fold lower in risk genotype cells. For rs2272165, chromatin marks and transcription factor binding were lower in the risk genotype, consistent with the above results. The risk allele (C) at rs6705628 increased promoter and enhancer activity across all three cell types. Consistently, rs6705628 chromatin marks and polymerase residency were significantly higher in the risk genotype. The observation from Western blots that DGUOK protein levels decrease in the risk-risk genotype suggests that the effects of the rs2272165 SNP perhaps outweigh those of the rs6705628 SNP.

Notably, ARID3A—which binds significantly more to the rs6705628 risk genotype—is broadly involved in the development and progression of SLE [37]. ARID3A expression levels in white blood cells correlated with disease activity in patients and were specifically linked to autoantibody production by B-cells. ARID3A expression also drives lineage fate in hematopoiesis, with high expression leading to greater numbers of B-cells, neutrophils, and plasmacytoid dendritic cells in SLE patients. The substantial binding of ARID3A specifically to the risk rs6705628 genotype (consistent with ENCODE data) may modulate downstream disease processes.

This locus was first reported in a meta-analysis, and the rs6705628 was flagged as a risk SNP for SLE in Chinese (odds ratio 0.71, *p* = 3 × 10^−8^) [16] and in Chinese and Thai (OR 0.75, *p* = 7 × 10^−17^) [15]. It was also noted as a risk locus for rheumatoid arthritis in East Asians and Europeans (OR 0.88, *p* = 7 × 10^−9^) [38]. Intriguingly, both risk alleles are the major and ancestral alleles; the derived non-risk alleles are very low abundance in European populations but quite common (~20%) in Asians, suggestive of positive selection for some desirable trait such as infection resistance. In fact, many SLE risk loci are targets of recent positive selection in this manner [39].

The two primary loci regulated by the two SNPs are *DGUOK* (mitochondrial deoxyguanosine kinase), which regulates mitochondrial DNA replication; and *DGUOK-AS1*, an antisense RNA directed against *DGUOK* [15]. *DGUOK* does not have an obvious immune system role and has been primarily studied in the context of mitochondrial DNA depletion and neurodevelopmental disorders [40]. Mitochondria are profoundly involved in the healthy function of the immune system (notably reactive oxygen generation and phagocytosis), though, and many of the most SLE-associated risk genes encode mitochondrial proteins [41], particularly components of mitochondrial NADPH oxidase [42,43], which produces reactive oxygen species. It is not unreasonable that alterations in the level of a critical mediator of mitochondrial DNA copy number and intactness would modulate oxidative phosphorylation (OxPhos), with knock-on effects on the immune system. The risk alleles are associated with lower DGUOK levels, potentially leading to less error correction in mitochondrial genomes, with potential downstream effects on other mitochondrial components. Indeed, SLE patients with high disease activity exhibit substantially decreased mitochondrial genome copy number and increased sequence heteroplasmy [44], indicative of error-correction failures.

We found that the *DGUOK-AS1* antisense RNA targets several immune-relevant microRNAs—in addition to the *DGUOK* transcript itself. *DGUOK-AS1* targets miR-1-3p (which facilitates Th17 differentiation [45]), miR-138-5p (which promotes TNFα-induced apoptosis through PTEN/PI3K/Akt signaling [46]), miR-148a-3p and miR-148b-3p (the primary biomarker for lupus nephritis [47] and a potent PTEN activator), miR-151a-3p (another biomarker of lupus nephritis [48]), miR-653-5p (an miRNA active in Behçet’s syndrome inflammation [49]), and miR-876-3p (which modulates proliferation and apoptosis in lymphocytic leukemia through JNK signaling [50]). A separate search for non-miRNA genes targeted by *DGUOK-AS1* revealed bradykinin—a key inflammatory mediator—and the lncRNA *Xist*, the primary determinant of X-inactivation and autoimmune sex bias. The fact that *DGUOK-AS1* is predicted to bind to the three most prominent lupus nephritis biomarkers is quite suggestive, although further experiments will be required to validate these predictions and establish mechanisms connecting the function of these miRNAs to SLE and nephritis progression. Similarly, the roles of bradykinin and *Xist*, and their regulation by *DGUOK-AS1* and *DGUOK*, will require further study.

Taken together, we have conclusively demonstrated the association of two SNPs near *DGUOK/DGUOK-AS1* with SLE, established transcriptional regulatory mechanisms and protein-binding effects of the risk alleles, and laid out potential downstream signaling pathways, mediated primarily through mitochondrial OxPhos activity and phagocytosis, and separately through competition with the function of myriad immune-involved microRNAs and *Xist*, the principal determinant of X chromosome inactivation and autoimmune sex bias. In-depth mechanistic studies will further delineate these functional targets for SLE susceptibility.

## Figures and Tables

**Figure 1 genes-13-01016-f001:**
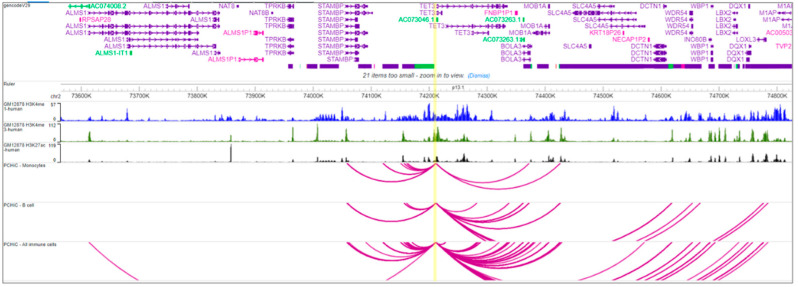
Chromatin interaction analysis using PCHiC showing the interacting region containing rs2272165 and promoters of neighboring target genes including *DGOUK* in GM12878 (B-lymphocytes) and monocytes. Chromatin interactions between the SNP (yellow vertical line) and target gene promoters are shown by magenta arcs.

**Figure 2 genes-13-01016-f002:**
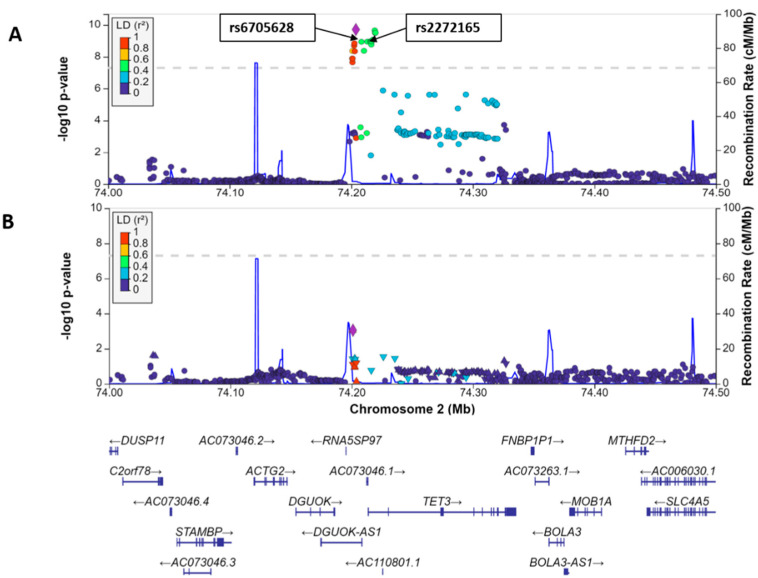
Ethnicity-specific meta-analysis, conditional analysis, and bioinformatics. (**A**) Plot shows −log_10_
*p*-value from the Asian cohorts on the left *y*-axis and physical position on the *x*-axis. The SNPs selected for conditional analysis are shown in boxes. Dots identify individual SNP *p*-values, colored by their LD strength (r^2^) with the lead SNP. Blue line indicates pre-calculated recombination rates (in cM/Mb, right y-axis) at each position. (**B**) SNP association after conditional analysis with the two selected SNPs.

**Figure 3 genes-13-01016-f003:**
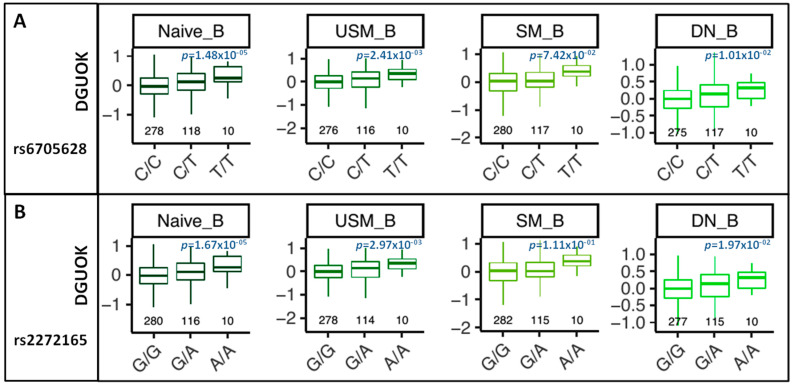
Immune cell type-specific eQTL analysis of the influence of rs6705628 (**A**) and rs2272165 (**B**) on target gene DGUOK (the only target gene) using transcriptome data from Asian samples. The genotypes and their corresponding sample sizes are shown at the bottom of each box plot. (C/C) is the risk for rs6705628 and (G/G) is the risk for rs2272165. USM = unstimulated, SM = stimulated, DN = double negative.

**Figure 4 genes-13-01016-f004:**
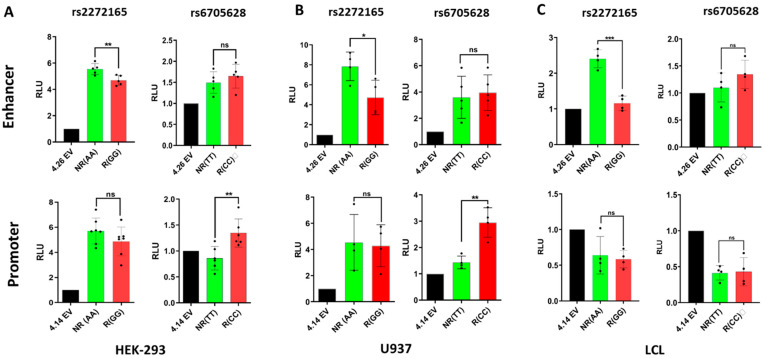
Allele-specific promoter and enhancer activity of the two SNPs, using luciferase reporter assays across (**A**) HEK293 (embryonic kidney cells), (**B**) U937 (monocyte), and (**C**) LCL GM18572 (B-lymphocytes). Empty vector pGL4.21 (EV) was used as reference. NR: non-risk and R: risk alleles; *p*-values are for Student’s *t*-test. * *p* < 0.05, ** *p* < 0.01, *** *p* < 0.001.

**Figure 5 genes-13-01016-f005:**
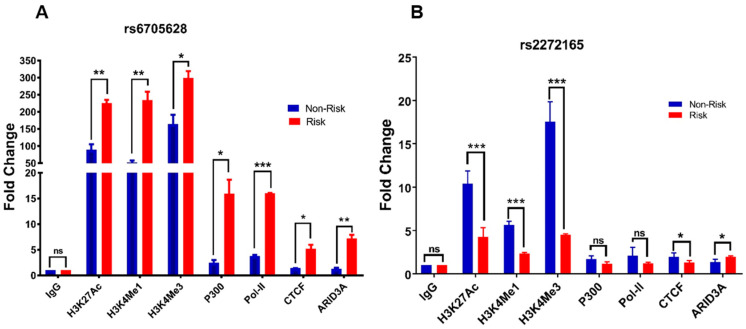
Allele-specific ChIP-qPCR of sequences containing SNPs (**A**) rs6705628 and (**B**) rs2272165 with histone marks and binding proteins (*x*-axis) using risk (GM18572) and non-risk (GM18553) LCL lines. *p*-values are for Student’s *t*-test. * *p* < 0.05, ** *p* < 0.01, *** *p* < 0.001.

**Figure 6 genes-13-01016-f006:**
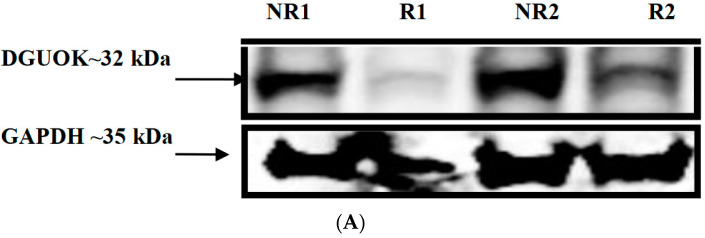
Western blot analysis of DGUOK protein in two risk (R1-GM18555, R2-GM18572) and two non-risk (NR1-GM18553, NR2-GM18561) genotype-specific LCLs (B-lymphoblastoid cells). DGUOK protein level was normalized to levels of glyceraldehyde 3-phosphate dehydrogenase (GAPDH). (**A**) Variation in DGUOK band intensity in genotype-specific LCLs. (**B**) Densitometry analysis showed that the normalized DGUOK expression was significantly reduced by ~3 fold in risk (R1 and R2; average R) compared to non-risk (NR1 and NR2; average NR) LCLs. *P*-value was determined by Student’s *t*-test. * *p* < 0.05.

**Table 1 genes-13-01016-t001:** Meta-analysis of rs6705628 and rs2272165 at 2p13.1 using Asian and European cohorts.

						rs6705628 (T/C)			rs2272165 (A/G)			
Ethnicity	Cohort	PMID	# Cases	# Controls	FU(A1)	FA(A1)	OR (95% CI)	*p*-Value	FU(A1)	FA(A1)	OR (95% CI)	*p*-Value
ASN	Meta-analysis		4144	11,014	0.19	0.16	0.79 (0.72–0.87)	5.56 × 10^–10^	0.19	0.16	0.79 (0.71–0.86)	3.48 × 10^−10^
	HC: Morris et al., 2016	27399966	1659	3398	0.18	0.15	0.72 (0.63–0.81)	3.23 × 10^−8^	0.18	0.15	0.71 (0.63–0.80)	1.86 × 10^−8^
	Korean: Sun et al., 2016	26808113	1710	6836	0.20	0.17	0.86 (0.77–0.961)	6.42 × 10^−3^	0.20	0.17	0.86 (0.77–0.96)	7.63 × 10^−3^
	HC: Sun et al., 2016	26808113	490	493	0.22	0.20	0.85 (0.67–1.07)	1.49 × 10^−1^	0.22	0.20	0.84 (0.67–1.06)	1.40 × 10^−1^
	Malaysian: Sun et al., 2016	26808113	285	287	0.17	0.11	0.66 (0.46–0.93)	1.36 × 10^−2^	0.17	0.11	0.67 (0.47–0.94)	1.72 × 10^−2^
EUR	Meta-analysis		6108	10,590	0.01	---	0.78 (0.55–1.0)	1.88 × 10^−2^	0.01	---	0.78 (0.55–1.0)	1.90 × 10^−2^
	EUR: Bentham et al., 2015	26502338	5201	9066	0.01	---	0.84 (0.65–1.08)	1.68 × 10^−1^	0.01	---	0.84 (0.65–1.08)	1.69 × 10^−1^
	SPN: Julia et al., 2018	29848360	907	1524	0.01	---	0.57 (0.31–1.04)	7.21 × 10^−2^	0.01	---	0.57 (0.31–1.04)	7.19 × 10^−2^

Abbreviations: FU: Minor allele frequency in unaffected individuals. FA: Minor allele frequency in affected individuals. OR: Odds Ratio. ASN: Asian ancestry. EUR: European ancestry.

## Data Availability

Publicly available datasets were analyzed in this study. The study references and PMIDs are shown in Table 1.

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
