# Peer review of "Discovery and Functional Characterization of Two Regulatory Variants Underlying Lupus Susceptibility at 2p13.1"

_genes, 2022, doi:10.3390/genes13061016_

Round 1
Reviewer 1 Report
Fazel-Najafabadi et al, provide an interesting study characterizing the role of two regulatory variants associated with SLE susceptibility. The authors performed a meta-analysis of four Asian and two European cohorts and identified ~20 variants that reached GWA significance (p<5x10-8). Further, they have used various functional assays and also exploited the publicly available resources to characterize the two variants rs6705628 and rs2272165 within TET3/DGUOK locus which were identified based on the conditional analysis.
Below are a few comments that the authors should address:
- Since the association signal in TET3 region is exclusive for the Asian population and not significant in European cohorts, the authors should clarify why meta-analysis is performed combing the two. It is unclear whether the 20 GWA significant SNPs are from the meta analysis of Asian cohorts alone or combined with European cohorts. In Supplementary Table1, the meta-pvalues are provided for Asian and European cohorts, while in the results section authors mention that they analyzed Asian and European ancestries both separately
and jointly, whereas the joint pvalues are not provided or mentioned. - Non-coding RNA interactions: Authors highlight that the lncRNA DGUOK-AS1 might act as a miRNA sponge. This observation is speculative and should be clarified in the results. Sponge effect can occur if the target gene has multiple binding site for a specific miRNA. For lncRNA-miRNA interactions, along with chromosome position and the analyzed region, binding type (k-mer), position of the miRNA interaction in the transcript, specific score, sequence of binding area and the numbers of conserved species, should be provided.
- The numbering of the Supplementary Tables should be corrected. In the manuscript Supp Table 2 and 3 are actually Table 3 and 4 in the supplementary files.
- Figures: All of the figures lack legends or description, making it very hard to interpret the results.
- Figure 1: Please mention if the meta analysis are from Asian cohorts alone or including both Asian and European cohorts
- Figure 4A: Please mention which target genes are being referred to in the figure title.
Author Response
Q1. (a) Since the association signal in TET3 region is exclusive for the Asian population and not significant in European cohorts, the authors should clarify why meta-analysis is performed combing the two.
Response: Thank you for pointing this out, and we are sorry that it was not clear before. Indeed, we did not perform a meta-analysis within the combined Asian and European populations. We now make clear that the meta-analysis was within each ethnicity in the text and the legend for Figure 1.
(b) It is unclear whether the 20 GWA significant SNPs are from the meta-analysis of Asian cohorts alone or combined with European cohorts.
Response: These 20 GWA significant SNPs are from an Asian-specific meta-analysis – now mentioned in Section 3.1.
(c) In Supplementary Table1, the meta-p values are provided for Asian and European cohorts, while in the results section authors mention that they analyzed Asian and European ancestries both separately
Response: In Supplementary Table 1, we provide the results of the ethnicity-specific meta-analysis separately for the Asian and European cohorts.
Q2. Non-coding RNA interactions: Authors highlight that the lncRNA DGUOK-AS1 might act as a miRNA sponge. This observation is speculative and should be clarified in the results. Sponge effect can occur if the target gene has multiple binding site for a specific miRNA. For lncRNA-miRNA interactions, along with chromosome position and the analyzed region, binding type (k-mer), position of the miRNA interaction in the transcript, specific score, sequence of binding area and the numbers of conserved species, should be provided.
Response: Thank you for these helpful comments. We have dramatically toned down this language in the Results, merely saying that it appears that DGUOK-AS1 might bind to multiple miRNA species and affect their function. Regarding the specific comments on the corresponding Supplementary Table, we are confused – these data come from DIANA-LncBase v3, which is an experimental – not computational – database. There has been no k-mer analysis performed – instead these data result from High-throughput sequencing of RNA isolated by crosslinking immunoprecipitation (HITS-CLIP) experiments. We now explicitly denote this in the table – the text already clearly stated that these were experimental findings and not computational predictions.
Q3. The numbering of the Supplementary Tables should be corrected. In the manuscript, Supp Table 2 and 3 are actually Table 3 and 4 in the supplementary files.
Response: Thank you for pointing this out. We are sorry for mislabelling the Supplementary Tables, and we have now corrected this in the manuscript.
Q4. Figures: All of the figures lack legends or descriptions, making it very hard to interpret the results.
Response: We are very sorry for this. We now include figure legends and descriptions.
- Figure 1: Please mention if the meta-analysis are from Asian cohorts alone or including both Asian and European cohorts
Response: See above.
- Figure 4A: Please mention which target genes are being referred to in the figure title.
Response: The target gene was DGUOK, now mentioned in the legend.

Reviewer 2 Report
Systemic lupus erythematosus is a complex, multisystem autoimmune condition, characterized by loss of self-tolerance with activation of autoreactive T and B cells leading to production of autoantibodies and tissue injury. The immune dysregulation is a consequence of interaction between genetic factors, environment and stochastic events. Genome-wide association studies revealed large number of lupus-associated risk loci, mostly linked to SNPs. However, there is a need to study potential risk variants in the functional context. Fazel-Najafabadi et al. indicated from 20 genome-wide significant variants two potential functional SNPs rs6705628 and rs2272165. This paper describe a complex functional analysis of SLE-associated polymorphisms and shed new light on disease mechanism, however there are points that must be addressed:
· Tables and figures have no legends, abbreviations need to be defined.
· Results of Western blot (Figure 5) are slightly unclear. How the protein level of DGUOK was assessed?
· Most of the figures and tables presented in the major text are duplicated in the supplementary data, making a bit difficult to find supplementary results.
Author Response
Q1. Tables and figures have no legends, abbreviations need to be defined.
Response: Thank you for pointing this out. This is now clearly written.
Q2. Results of Western blot (Figure 5) are slightly unclear. How the protein level of DGUOK was assessed?
Response: We are sorry for this. We now clearly describe this in the legend to Figure 5.
Q3. Most of the figures and tables presented in the major text are duplicated in the supplementary data, making a bit difficult to find supplementary results.
Response: Sorry for that, and we will double-check this before uploading the files.
